# Impact of Primary Hemostasis Disorders on Late Major Bleeding Events among Anticoagulated Atrial Fibrillation Patients Treated by TAVR

**DOI:** 10.3390/jcm11010212

**Published:** 2021-12-31

**Authors:** Laurent Dietrich, Marion Kibler, Kensuke Matsushita, Benjamin Marchandot, Antonin Trimaille, Antje Reydel, Bamba Diop, Phi Dinh Truong, Anh Mai Trung, Annie Trinh, Adrien Carmona, Sébastien Hess, Laurence Jesel, Patrick Ohlmann, Olivier Morel

**Affiliations:** 1Centre Hospitalier Universitaire, Pôle d’Activité Médico-Chirurgicale Cardio-Vasculaire, Nouvel Hôpital Civil, Université de Strasbourg, 67000 Strasbourg, France; laurent_dietrich@yahoo.com (L.D.); matsuken_22@yahoo.co.jp (K.M.); benjaminmarchandot@gmail.com (B.M.); antonin.trimaille@chru-strasbourg.fr (A.T.); anne-claire.reydel@chru-strasbourg.fr (A.R.); bamba.diop@chru-strasbourg.fr (B.D.); annie.trinh@chru-strasbourg.fr (A.T.); adrien.carmona@chru-strasbourg.fr (A.C.); sebastien.hess@chru-strasbourg.fr (S.H.); laurence.jesel@chru-strasbourg.fr (L.J.); patrick.ohlmann@chru-strasbourg.fr (P.O.); olivier.morel@chru-strasbourg.fr (O.M.); 2Institut National de la Santé et de la Recherche Médicale (INSERM), Nano Médecine Régénérative, Unité Mixte de Recherche 1260, Faculté de Pharmacie, Université de Strasbourg, 67400 Illkirch, France; 3Vietnam National Heart Institute, Bach Mai Hospital, Hanoi 100000, Vietnam; drphi.vtm@gmail.com (P.D.T.); trung-anh.mai@chru-strasbourg.fr (A.M.T.)

**Keywords:** transcatheter aortic valve replacement, primary hemostasis disorders, closure time with ADP, major life/threatening bleedings

## Abstract

Background: Bleeding events are among the striking complications following transcatheter aortic valve replacement (TAVR), and bleeding prediction models are crucially warranted. Several studies have highlighted that primary hemostasis disorders secondary to persistent loss of high-molecular-weight (HMW) multimers of the von Willebrand factor (vWF) and assessed by adenosine diphosphate closure time (CT-ADP) may be a strong predictor of late major/life-threatening bleeding complications (MLBCs). Pre-existing atrial fibrillation (AF) is a frequent comorbidity in TAVR patients and potentially associated with increased bleeding events after the procedure. Objectives: This study evaluated the impact of ongoing primary hemostasis disorders, as assessed by post-procedural CT-ADP > 180 s, on clinical events after TAVR among anticoagulated AF patients. Methods: An ongoing primary hemostasis disorder was defined by post-procedure CT-ADP > 180 s. Bleeding complications were assessed according to the Valve Academic Research Consortium-2 (VARC-2) criteria. The primary endpoint was the occurrence of late MLBCs at one-year follow-up. The secondary endpoint was a composite of mortality, stroke, myocardial infarction, and rehospitalization for heart failure. Results: In total, 384 TAVR patients were included in the analysis. Of these patients, 57 patients (14.8%) had a prolongated CT-ADP > 180 s. Increased MLBCs were observed in patients with CT-ADP > 180 s (35.1% versus 1.2%; *p* < 0.0001). Conversely, the occurrence of the composite endpoint did not differ between the groups. Multivariate analysis identified CT-ADP > 180 s (HR 28.93; 95% CI 9.74–85.95; *p* < 0.0001), bleeding history, paradoxical aortic stenosis (AS), and major vascular complications following TAVR as independent predictors of late MLBCs. Conclusion: Among patients with anticoagulated AF, a post-procedural CT-ADP > 180 s was identified as a strong independent predictor of late MLBCs. These findings suggest that persistent primary hemostasis disorders contribute to a higher risk of late bleeding events and should be considered for a tailored, risk-adjusted antithrombotic therapy after TAVR.

## 1. Introduction

Whilst significant advances in transcatheter aortic valve replacement (TAVR) device technology have enabled a drastic reduction of periprocedural bleeds and intraoperative complications, the rates of late bleedings events after TAVR are still consistent over time and have been associated with increased mortality [1,2,3,4,5,6].

Despite continuous progress in techniques and methods to minimize periprocedural complications including paravalvular leak or bleeds [7,8,9], several studies have suggested that primary hemostasis disorders secondary to persistent loss of high-molecular-weight (HMW) multimers of the von Willebrand factor (vWF) may still contribute to late bleeding complications [2,10]. vWF is a multimeric protein secreted by endothelial cells and platelets. This HMW glycoprotein mediates the adhesion of platelets to sites of vascular damage through glycoprotein Ib-vWF interactions [11].

Closure time with adenosine diphosphate (CT-ADP) as assessed with the platelet function analyzer PFA-100 (Siemens Healthcare Diagnostics, Marburg, Germany) is a primary hemostasis point-of-care test used as a surrogate marker of HMW multimer defects. CT-ADP assessed by PFA-100 has proven to be highly sensitive to HMW multimer defects [12,13]. Persistent prolongation of CT-ADP (>180 s) after TAVR was highlighted as a strong predictor of late major/life-threatening bleeding complications (MLBCs) [10] and was predictive of increased mortality one year after the procedure [13].

Given the advanced age and the presence of multiple co-morbidities including atrial fibrillation (AF) in the TAVR population, late bleeding events, particularly in the context of anticoagulation, may be frequent and may have a negative effect on the long-term prognosis [1,3]. The reported prevalence of pre-existing AF in patients undergoing TAVR ranges from 16% to 51% [14,15], and 50% of these patients were anticoagulated [16]. Recent studies have shown an increase in early and late mortality in patients with pre-existing AF [14,17,18]. Hemodynamic injury, renal failure, thrombo-embolic events, and bleeding were proposed as key determinants of adverse outcomes in AF patients [19,20,21].

However, the interaction between pre-existing AF and MLBCs is still a matter of debate. Whilst the reports by Généreux et al. [1] and Tarantini et al. [17] highlighted an enhanced bleeding risk in AF patients, other studies failed to evidence a noxious association between AF and bleeds [18,19]. Altogether, these findings suggest that specific mechanisms leading to hemorrhagic complications in this population have not been fully characterized. The question of whether ongoing loss of HMW multimers in addition to anticoagulation could promote MLBCs remains unsolved.

In the present study, we aimed to evaluate the impact of ongoing primary hemostasis disorders, as assessed by CT ADP > 180 s on MLBCs among patients with anticoagulated AF after TAVR.

## 2. Materials and Methods

### 2.1. Population

A total of 1125 patients undergoing TAVR were enrolled in a prospective registry at our institution (Nouvel Hôpital Civil, University of Strasbourg, Strasbourg, France) between February 2010 and May 2019. Indications for TAVR and procedural approaches were assessed by the local heart team composed of interventional cardiologists and cardiac surgeons. The logistic EuroSCORE, as well as other non-traditional risk factors such as non-cardiac comorbidities or frailty were considered when determining patients’ specific risk profiles. All patients signed informed consent forms before the procedure and agreed to the anonymous processing of their data (FRANCE 2 and FRANCE TAVI Registries).

The TAVR procedures were performed using balloon-expandable Edwards SAPIEN XT or S3 prosthesis (Edwards Lifesciences, Irvine, CA, USA) devices and self-expandable Core Valve or Evolut R (Medtronic, Minneapolis, MN, USA) devices.

Patients received intravenous unfractionated heparin to achieve an activated clotting time of 250 s to 350 s during the procedures. The standard heparin dosage was used during the procedure. Heparin was antagonized with 100 IU/kg of protamine at the end of the procedure.

All patients were treated with oral anticoagulant (OAC), and a dual therapy including OAC and aspirin was introduced and continued for 1–3 mo after TAVR, then OAC was maintained as a monotherapy. The exclusion criteria included periprocedural death, no history of AF or new onset AF, non-anticoagulated AF, and the absence of CT-ADP assays.

### 2.2. Collection of Data

All baseline and follow-up variables were recorded and entered into a secure database. The aim of this study was to identify the prognostic impact of CT-ADP > 180 s on the clinical outcomes of anticoagulated AF patients after TAVR. The primary endpoint was the occurrence of MLBCs > 30 d after TAVR and censored at 1 y. The secondary endpoint was a composite endpoint defined by mortality, stroke, myocardial infraction, and rehospitalization for heart failure. In addition, we sought to characterize the determinants of post-procedural ongoing hemostasis disorders.

All patients were contacted by phone to ascertain their health status, cardiovascular symptoms, and outcomes. Patient-reported data collected through a standardized questionnaire were thoroughly cross-checked with official clinical records.

In the case of several hemorrhagic events, only the first event was considered in the statistical analysis.

### 2.3. Blood Collection

Whole-blood samples were collected by venipuncture the day before and 24 h after TAVR. Analysis of CT-ADP with the primary hemostasis point-of-care assay PFA-100 (Siemens Healthcare, Marburg, Germany) was performed according to the manufacturer’s recommendation. In accordance with previous works [12,13], this measurement was performed as a surrogate marker of HMW multimers of vWF defects, which was not directly measured by electrophoresis. We have described that a threshold value for post-procedural CT-ADP of more than 180 s could be an independent predictor of late MLBCs [10]. In addition, the extent of P2Y12 inhibition by clopidogrel was evaluated by the analysis of VASP phosphorylation by flow cytometry as previously described [16].

### 2.4. Paravalvular Aortic Leak Evaluation

Following the TAVR procedure, a significant paravalvular leak (PVL) was defined by transthoracic echocardiography as the circumferential extent of regurgitation > 10% (more than mild) according to the definition provided by the Valve Academic Research Consortium-2 (VARC-2). Color Doppler evaluation was performed just below the valve stent for paravalvular jets.

### 2.5. Clinical Endpoints

Bleeding complications were assessed according to VARC-2 criteria with slight modifications and were classified as follows: life-threatening bleeding, major bleeding, and minor bleeding. Major or life-threatening bleeding occurring before the first 30 d after TAVR was categorized under early MLBCs. We added to the VARC-2 classification the occurrence of non-exteriorization bleeding requiring red blood cell transfusion ≥2 units. Late MLBCs were defined as major or life-threatening bleeding events occurring 1–12 mo after TAVR. Myocardial infarction (MI) was defined according to the fourth universal definition of MI [22], and stroke was defined according to the definition of the American Heart and Stroke Association [23]. Rehospitalization for heart failure was defined as any rehospitalization requiring the administration of intravenous diuretic therapy.

### 2.6. Statistical Analysis

Descriptive statistics are expressed as the means ± the standard deviations, or the median (25th–75th) for continuous data, or as counts (percentages) for categorical variables. Logistic regression models were constructed to evaluate the unadjusted and covariate-adjusted hazards ratio (HR) and 95% confidence interval (CI) for the predictors of late MLBCs and CT-ADP > 180 s. Variables adjusted for in the multivariate models were those showing univariate associations at a *p*-value < 0.10. Survival curves according to the CT-ADP were plotted by the Kaplan–Meier method (log-rank test) with right censoring at the time of last follow-up (8 May 2020). The time-to-event was calculated as the time elapsed from TAVR to the date of the MLBCs. Statistical analyses were performed using SPSS, Version 17.0 (IBM, Armonk, NY, USA). All tests were two-sided, and statistical significance was set as a *p*-value < 0.05.

## 3. Results

### 3.1. Demographics

Between 2010 and 2019, a total of 1125 patients with aortic stenosis (AS) underwent TAVR at our institution (Figure 1). Eleven (0.98%) patients died during the procedure, and twenty patients were lost to follow-up. Overall, 546 (48.5%) patients without a history of AF or new onset AF, 92 (8.2%) patients with non-anticoagulated AF, and 72 (6.4%) patients without available CT-ADP value were excluded from the analysis. The remaining 384 included patients were followed for a median duration of 545 d (257 25th–730 75th). Of these, 355 patients (92.4%) had a transfemoral approach and 21 patients (5.4%) had a carotid access. Among them, post-procedural CT-ADP > 180 s was evidenced in 57 patients (14.8%).

Baseline, procedural, biological, and post-procedural characteristics according to a CT-ADP of 180 s are displayed in Table 1.

Coronary artery disease, arterial hypertension, peripheral artery disease, dialysis, and dual-antiplatelet therapy (DAPT) were more frequently represented in the CT-ADP > 180 s group. Baseline echocardiographic characteristics were similar between the two subsets.

As expected, balloon post-dilatation was more frequently observed in the CT-ADP > 180 s group. Likewise, higher rates of significant PVL before discharge and one month after TAVR were observed in the CT-ADP > 180 s group. No differences between the type of anticoagulant treatment (vitamin K antagonist (VKA) or direct oral anticoagulant (DOAC)) could be established. Moreover, P2Y12 inhibition, as assessed by PRI-VASP, was equivalent among the groups.

### 3.2. Impact of Primary Hemostasis Disorders on Clinical Events

At 1 y follow-up, late MLBCs was evidenced in 24 patients (6.3%). The types of late MLBCs are listed in Table 2. Gastrointestinal (GI) bleeding (n = 17 (70.8%)) was the most frequent type of late MLBC. Muscular-cutaneous bleeding was the second-most frequent type of late MLBCs, occurring in five (20.8%) patients. Neurological and genitourinary bleedings were rare, both identified in one (4.2%) patient.

The rate of late MLBCs was drastically elevated in patients with ongoing primary hemostasis disorders (35.1% versus 1.2%, respectively; *p* < 0.0001; Table 3). Likewise, and consistent with a bleeding propensity, periprocedural bleeds and transfusion were more frequently observed in patients with CT-ADP > 180 s. Conversely, the rates of secondary endpoint occurrence did not significantly differ among the groups.

### 3.3. Predictors of Late MLBCs

In the univariate analysis, dialysis, bleeding history, paradoxical low-flow, low-gradient AS, hemoglobin, CT-ADP > 180 s, and major vascular complications after TAVR were significant predictors of late MLBCs. Conversely, no impact of antithrombotic treatment including DAPT or the extent of P2Y12 inhibition on the bleeding risk was observed.

In the multivariate analysis, bleeding history (HR 2.72; 95% CI 1.07–6.90; *p* = 0.035), paradoxical low-flow, low-gradient AS (HR 4.35; 95% CI 1.49–12.71; *p* = 0.007), CT-ADP > 180 s after TAVR (HR 28.93; 95% CI 9.74–85.95; *p* < 0.0001), and major vascular complications after the TAVR procedure (HR 3.01; 95% CI 1.12–8.10; *p* = 0.029) were identified as independent predictors of late MLBCs (Table 4).

Kaplan–Meier plots of late MLBC-free survival according to the CT-ADP threshold > 180 s are represented in Figure 2.

### 3.4. Predictors of CT-ADP > 180 s after TAVR

Additional analyses were performed to identify predictors of CT-ADP > 180 s and potential confounding factors. In the univariate analysis, stage V CKD under dialysis, history of coronary artery disease, left ventricular mass index (LVMi) assessed by echocardiography, pre-procedural hemoglobin level and platelet count, DAPT, mean aortic gradient, and significant PVL were associated with a prolongated CT-ADP > 180 s after TAVR.

According to the multivariate analysis, dialysis (HR 6.45; 95% CI 1.05–39.49; *p* = 0.044), LVMi (HR 1.01; 95% CI 1.00–1.02; *p* = 0.033), baseline anemia, thrombopenia (HR 0.76; 95% CI 0.62–0.95; *p* = 0.013 and HR 0.99; 95% CI 0.98–1.00; *p* = 0.002, respectively), and significant PVL (HR 2.74; 95% CI 1.26–5.94; *p* = 0.011) were evidenced as independent predictors of prolongated CT-ADP > 180 s after TAVR. No significant association between prolongated CT-ADP > 180 s and antithrombotic regimen was identified (Table 5).

## 4. Discussion

The current report is the first to specifically investigate the impact of primary hemostasis disorders as assessed by CT-ADP > 180 s on late bleeding events in anticoagulated AF patients.

The salient results of the present study are as follows: (1) Late major and life/threatening bleedings after TAVR occurred in 6.3% of patients with anticoagulated AF. In this specific population, patients with a CT-ADP > 180 s are 28.9-times more likely to experience late MLBCs than patients with a CT-ADP ≤ 180 s. (2) CT-ADP > 180 s was an independent and strong predictor of late MLBCs 1 y after TAVR. (3) Post-procedural significant PVL, LV hypertrophy, CKD on dialysis, baseline anemia, and thrombopenia were associated with prolonged CT-ADP values and should be considered when analyzing CT-ADP.

Altogether, these findings suggest that persistent primary hemostasis disorders following TAVR are easily assessed by the CT-ADP assay and contribute to a higher risk of late bleeding complications among patients with anticoagulated AF.

### 4.1. Late MLBCs after TAVR

Numerous studies have stressed the high frequency of late MLBCs after TAVR and its deleterious impact on patient outcomes, including enhanced mortality [1,2,10]. In the present study, derived from a homogeneous population of AF treated with anticoagulants, late MLBCs mainly of GI origin (70.8%) occurred in 6.3 % of patients 1 y after TAVR. In the seminal work by Généreux et al. derived from a large cohort of 2401 patients enrolled in the PARTNER trial [1], late MLBCs occurred in 5.9% of patients 1 y after TAVR and were associated with a four-fold increase in late mortality. In this report, PVL, a condition associated with ongoing HMW proteolysis, as well as baseline hemoglobin level and AF were identified as important predictors of MLBCs. The question of whether AF per se impacts late bleeding risk remains a debate. In the Bern registry, no impact of AF on MLBCs could be established. Likewise, in the large FRANCE 2 registry, pre-existing and new onset AF were associated wither higher mortality after TAVR, but no specific impact of AF and late bleedings could be established. In this registry, the incidence of MLBCs was 10.4% at 1 y follow-up in 1002 AF patients and comparable to that observed in 2622 (11.1%) non-AF patients [19]. However, in the absence of the systematic capture of antithrombotic regimens in the PARTNER studies or FRANCE 2 registry, the respective contribution of AF or a concomitant anticoagulant regimen on late bleedings could not be precisely delineated.

### 4.2. Impact of Ongoing Primary Hemostasis Disorders on Clinical Events

Among various factors that could promote late bleeds in AF patients, we carefully considered the role played by primary hemostasis disorders and acquired von Willebrand syndrome. Severe AS induces highly turbulent blood flow at the vicinity of the aortic valve, causing high shear stress that promotes deployment and cleavage of vWF HMW multimers. Several groups, including ours, have demonstrated that TAVR could efficiently resolve HMW multimer defects within minutes and normalize primary hemostasis dysfunction. When significant PVLs occur, flow turbulence persists after valve replacement and may contribute to increased bleeding risk. In the present study focusing on AF patients, post-procedural prolonged CT-ADP was evidenced in a substantial proportion of patients (14.8%) and was associated with a dramatic increase of late MLBCs, but also periprocedural complications and transfusion need. In addition, we identified that elevated CT-ADP values were associated with low hemoglobin levels, low platelet count, stage V CKD under dialysis, and elevated LVMi, suggesting that CT-ADP can reflect a broad spectrum of pathologies possibly acting as potential confounders. In contrast, no significant impact of the type of antithrombotic regimen nor P2Y12 inhibition could be established on the bleeding risk. Several studies have emphasized the link between HMW multimer defects, angiodysplasia, and GI bleeds, and vicious amplification loops associated with the loss of vWF, dysfunctional angiogenesis, and bleeding were recently described. Challenging the sole role of prolonged CT-ADP as a marker of bleeding risk, other studies have suggested that prolonged CT-ADP could be associated with higher mortality [13] or increased rates of ischemic cerebrovascular events [24]. In the latter study, a noxious relationship among ongoing primary hemostasis disorders, early bleeds, transfusion, and ischemic stroke was evidenced. In the present study, no significant impact of prolonged CT-ADP on mortality was evidenced. Whilst the description of the precise mechanisms linking prolonged CT-ADP and adverse outcome is beyond the scope of the present study, it is likely that the drastic reduction of PVL over time as the combined consequence of technologic refinement and cumulate experiences has blunted the impact of ongoing primary hemostasis disorders on other endpoints such as heart failure or cardiovascular mortality [25]. In our experience, other significant predictors of late MLBCs are represented by bleeding history, major vascular complications, and paradoxical low-flow, low-gradient AS. Recent studies and meta-analyses have emphasized the frailty of the TAVR population (older age, female predominance, higher prevalence of hypertension), which is associated with adverse outcomes after TAVR [26,27,28].

### 4.3. CT-ADP and Bleeding Risk Management after TAVR in AF Patients

Thus far, the prevention of thrombotic events after TAVR in AF patients has relied on lifelong OAC regardless of the bleeding risk, with optional administration of aspirin or clopidogrel [29]. In defining optimal antithrombotic strategies, an important hint was recently provided by the POPULAR TAVI trial cohort B [30]. In this study, patients undergoing TAVR with an indication of OAC (94.9% of AF patients) were randomized to receive either OAC alone or OAC plus clopidogrel for 3 mo. OAC alone was associated with a reduction in bleeding events, including a two-fold decrease of late MLBCs without impacting the rate of major adverse ischemic outcomes. In the POPULAR TAVI trial cohort B [30], most bleeding events occurred in the first weeks after the procedure, suggesting a determinant role of clopidogrel administration. Regardless of AF, the initial enthusiasm for the liberal use of OACs in TAVR was dampened by the early discontinuation of the GALILEO study [31]. In this trial, the efficacy and tolerance of rivaroxaban 10 mg/d combined with aspirin 75 mg/d for 3 mo compared with conventional DAPT for three months showed an increase in all-cause mortality, thromboembolic events, and bleeding events in the rivaroxaban group. Whilst the prevention of thrombotic risk is of paramount importance in AF patients, our study clearly highlighted that the therapeutic window of anticoagulant use is narrow in patients with ongoing primary hemostasis disorders characterized by a dramatic enhancement of late MLBCs. In our experience, the analysis of the Kaplan–Meier plots of late MLBC-free survival according to the CT-ADP > 180 s showed that the onset of late MLBCs was significantly continuous throughout the first year of follow-up in cases of CT-ADP > 180 s, despite the planned withdrawal of aspirin between 1 mo and 3 mo after the procedure. Moreover, the type of antithrombotic regimen (DOAC or VKA associated with aspirin, DAPT associated with OAC) or the extent of P2Y12 inhibition did not appear to significantly contribute to late MLBCs. From a pragmatic approach, the present data clearly demonstrated the existence of a noxious relationship between ongoing primary hemostasis disorders and late bleeding events in anticoagulated AF patients. In line with this paradigm, strategies aimed at reducing late MLBCs in patients under OAC represent an important area of patient care improvement. Further trials are needed to assess the potential benefit of a more restrictive anti-thrombotic treatment in this vulnerable population with ongoing primary hemostasis disorders such as reduced dosing of anticoagulation after the procedure to minimize the bleeding risk without increasing the risk of thrombotic events, planned appendage closure, or the systematic use of proton pump inhibitors [32]. Routine assessment of CT-ADP after TAVR could be viewed as a reliable tool to tailor post-TAVR antithrombotic and anticoagulation therapy.

### 4.4. Study Limitations

Several caveats of our investigation must be considered: (1) The direct measurement of HMW vWF defects was not performed. (2) CT-ADP prolongation is not specific for HMW multimer vWF defects and could be influenced by various other parameters (e.g., platelet count, hemoglobin level, and AS type). (3) CT-ADP was only performed 24 h after TAVR and was not repeated during the follow-up at the time of the bleeding event. (4) DAPT was significantly higher in the CT-ADP > 180 s group, but DAPT and VASP after TAVR were not described as independent factors of late MLBCs and prolongated CT-ADP > 180 s. (5) The antithrombotic regimen at the time of the bleeding event was not captured. (6) The number of observed MLBCs was limited. As such, the presence of residual confounders may pose limitations to the generalizability of our conclusions.

## 5. Conclusions

Among patients with anticoagulated AF, a post-procedural CT-ADP > 180 s as a surrogate marker of HMW multimer defects is an independent predictor of late MLBCs1 y after TAVR. These findings suggest that persistent primary hemostasis disorders contribute to bleeding diathesis and may be considered for a better individualized, risk-adjusted antithrombotic therapy after TAVR. Further trials are needed to evaluate tailored antithrombotic strategies enabling a better compromise between ischemic and bleeding risk in patients with ongoing primary hemostasis disorders.

## Figures and Tables

**Figure 1 jcm-11-00212-f001:**
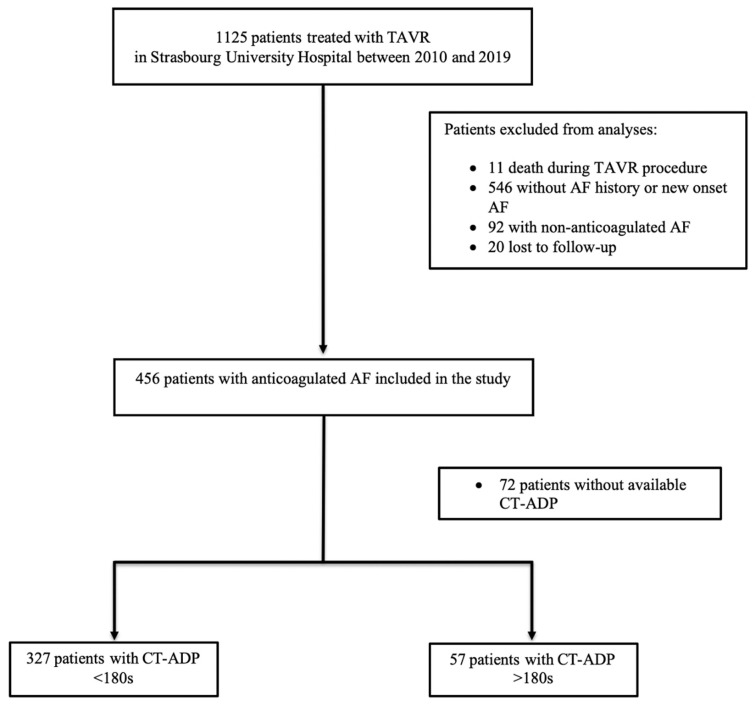
Flowchart of the study. ADP = adenosine diphosphate; AF = atrial fibrillation; CT-ADP = closure time with ADP; TAVR = transcatheter aortic valve replacement.

**Figure 2 jcm-11-00212-f002:**
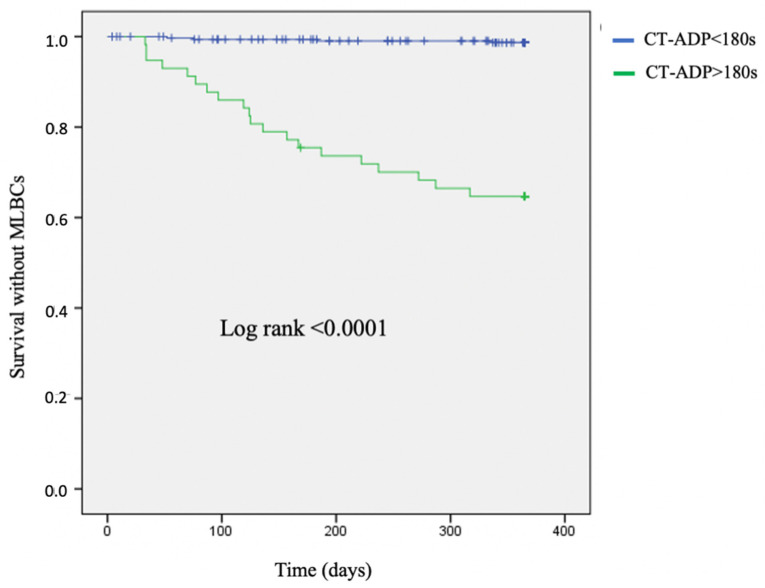
Kaplan–Meier curves. Survival without late major or life-threatening bleeding complications (MLBCs) according to a post-procedural CT-ADP threshold of > 180 s. ADP = adenosine diphosphate; CT-ADP = closure time with ADP; MLBCs = major or life-threatening bleeding complications.

**Table 1 jcm-11-00212-t001:** Baseline, procedural, post-procedural, and biological characteristics.

		AF + CT-ADP ≤ 180 s (*n* = 327)	AF + CT-ADP > 180 s (*n* = 57)	*p*-Value
Baseline Characteristics				
Age—y		83.2 ± 6.3	83.9 ± 6.4	0.320
Male sex—no (%)		158 (48.3)	29 (50.9)	0.415
Hypertension—no (%)		271 (82.9)	53 (93.0)	0.034
Diabetes—no (%)		100 (30.6)	23 (40.4)	0.097
BMI—kg/m^2^		27.9 ± 6.2	26.1 ± 4.6	0.360
EuroSCORE—%		19.5 ± 12.6	24.3 ± 18.6	0.824
CKD—no (%)		68 (20.8)	15 (26.3)	0.221
Dialysis—no (%)		6 (1.8)	4 (7.0)	0.046
COPD—no (%)		55 (16.8)	4 (7.0)	0.038
Coronary artery disease—no (%)		142 (43.4)	34 (59.6)	0.017
Stroke—no (%)		52 (15.9)	11 (19.3)	0.319
Peripheral artery disease—no (%)		102 (31.2)	11 (19.3)	0.045
Bleeding history—no (%)		36 (11.0)	8 (14.0)	0.320
Peptic ulcer disease or gastrointestinal bleeding—(%)		13 (4.0)	4 (7.0)	0.235
Normal-flow, high-gradient AS—no (%)		256 (78.3)	43 (75.4)	0.372
Low-flow, low-gradient AS—no (%)		45 (13.8)	6 (10.5)	0.337
Paradoxical low-flow, low-gradient AS—no (%)		20 (6.1)	5 (8.8)	0.307
LVEF—%		51.5 ± 15.4	51.6 ± 17.5	0.486
LVMi—g/m^2^		131.0 ± 40.2	147.5 ± 31.3	0.263
Mean aortic valve gradient—mm Hg		44.5 ± 12.7	42.7 ± 12.5	0.651
Aortic valve area—cm^2^		0.75 ± 0.21	0.75 ± 0.33	0.120
SPAP—mmHg		43.7 ± 14.1	44.2 ± 16.4	0.772
CT aortic annulus	Diameter—mm	25.0 ± 3.2	25.8 ± 3.4	0.247
	Area—mm^2^	495.3 ± 115.5	505.5 ± 125.0	0.400
Hb—g/dL		12.2 ± 1.7	11.6 ± 1.9	0.088
Platelet count—/mm^3^		231.4 ± 72.9	198.6 ± 56.7	0.673
CT-ADP—s		158.6 ± 66.2	201.7 ± 76.4	0.602
Creatinine—umol/L		118.7 ± 65.8	142.7 ± 95.7	0.642
DAPT—no (%)		33 (10.1)	12 (21.1)	0.020
Prior balloon aortic valvuloplasty—no (%)		17 (5.2)	4 (7.0)	0.380
Procedural Characteristics				
Approach	Transfemoral—no (%)	300 (91.7)	55 (96.5)	0.163
	Carotid access—no (%)	20 (6.1)	1 (1.8)	0.152
Valve	SAPIEN—no (%)	198 (60.6)	35 (61.4)	0.513
	Core Valve—no (%)	129 (39.4)	22 (38.6)	0.487
Balloon reimpaction—no (%)		35 (10.7)	13 (22.8)	0.014
Post-Procedural Characteristics				
Antithrombotic regimen	DAPT—no (%)	31 (9.5)	11 (19.3)	0.030
	Aspirin—no (%)	303 (92.7)	50 (87.7)	0.158
	Clopidogrel—no (%)	37 (11.3)	11 (19.3)	0.076
	VKA—no (%)	190 (58.1)	37 (64.9)	0.207
	DOAC—no (%)	131 (40.1)	18 (31.6)	0.143
Hb– g/dL		10.3 ± 1.4	10.0 ± 1.2	0.071
Platelet count—/mm^3^		128.8 ± 79.6	177.9 ± 169.5	0.077
CT-ADP—s		111.4 ± 27.6	265.6 ± 45.9	<0.0001
PRI-VASP—%		74.4 ± 13.2	71.6 ± 14.9	0.722
Creatinine—umol/L		72.4 ± 57.4	64.8 ± 65.7	0.151
LVEF—%		54.7 ± 12.5	54.4 ± 12.9	0.672
Mean aortic gradient—mmHg		7.3 ± 4.1	8.5 ± 4.7	0.217
Significant PVL—no (%)		30 (9.2)	22 (38.6)	< 0.0001
Significant PVL one month after TAVR—no (%)		21 (6.4)	15 (26.3)	< 0.0001
LVEF at 1 year—%		56.5 ± 11.6	60.7 ± 9.3	0.114
Mean aortic gradient at 1 year—mmHg		10.3 ± 7.3	10.3 ± 5.3	0.565

Values are no (%) or mean SD. ADP = adenosine diphosphate; AF = atrial fibrillation; AS = aortic stenosis; BMI = Body Mass Index; COPD = chronic obstructive pulmonary disease; CKD = chronic kidney disease (creatinine > 150mol/L); CRP = C-reactive protein; CT = computed tomography; CT-ADP = closure time with ADP; DAPT = dual-antiplatelet therapy; DOAC = direct-acting oral anticoagulant; EuroSCORE = logistic EuroSCORE predicted risk of mortality at 30 d; Hb = hemoglobin level; HTA = arterial hypertension; LVEF = left ventricular ejection function; LVMi = left ventricular mass index; OAC = oral anticoagulation; PRI-VASP = platelet reactivity index VASP; SPAP = systolic pulmonary arterial pressure; PVL = paravalvular aortic regurgitation; TAVR = transcatheter aortic valve replacement; VKA = vitamin K antagonist; WBC = white blood cells.

**Table 2 jcm-11-00212-t002:** Type of late MBLCs.

Type of Late MLBCs	Late MLBCs Events	Rate (%)
Gastro-intestinal	17	70.8
Muscular-cutaneous	5	20.8
Neurological	1	4.2
Genitourinary	1	4.2

Abbreviations are the same as Table 1.

**Table 3 jcm-11-00212-t003:** Impact of atrial fibrillation and CT-ADP > 180 s on clinical outcomes.

		AF + CT-ADP ≤ 180 s (*n* = 327)	AF + CT-ADP > 180 s (*n* = 57)	*p*-Value
Late MLBCs—no (%)		4 (1.2)	20 (35.1)	<0.0001
Composite endpoint—no (%)		163 (49.8)	35 (61.4)	0.071
Death—no (%)		122 (37.3)	25 (43.9)	0.214
Cardiovascular death—no (%)		49 (15.0)	10 (17.5)	0.376
Hospitalization for heart failure—no (%)		73 (22.3)	18 (31.6)	0.091
Ischemic stroke—no (%)		25 (7.6)	4 (7.0)	0.564
Hemorrhagic stroke—no (%)		2 (0.6)	1 (1.8)	0.383
Myocardial infarction—no (%)		5 (1.5)	2 (3.5)	0.279
Transfusion immediately after TAVR—no (%)		52 (15.9)	19 (33.3)	0.003
Vascular complications after TAVR	Major—no (%)	28 (8.6)	9 (15.8)	0.077
	Minor—no (%)	54 (16.5)	13 (22.8)	0.166
Early bleeding after TAVR	Major + Life threatening—no (%)	45 (13.8)	21 (36.8)	<0.0001
	Minor—no (%)	52 (15.9)	11 (19.3)	0.319

Values are no (%). All percentages correspond to the 1 y cumulative incidence. ADP = adenosine diphosphate; AF = atrial fibrillation; CT-ADP = closure time with ADP; MLBCs = major or life-threatening bleeding complications; TAVR = transcatheter aortic valve replacement.

**Table 4 jcm-11-00212-t004:** Predictors of late MLBCs.

	Univariate Analysis			Multivariate Analysis		
	HR	95% CI	*p*-Value	HR	95% CI	*p*-Value
Baseline Characteristics						
Hypertension	1.32	0.39–4.43	0.652			
CKD	1.27	0.50–3.19	0.618			
Dialysis	6.73	2.00–22.58	0.002	2.53	0.64–10.04	0.188
Coronary artery disease	1.39	0.62–3.10	0.425			
Peripheral artery disease	0.22	0.05–0.92	0.039			
Bleeding history	3.41	1.41–8.22	0.006	2.72	1.07–6.90	0.035
Normal-flow, high-gradient AS	0.67	0.28–1.62	0.377			
Low-flow, low-gradient AS	0.28	0.04–2.04	0.208			
Paradoxical low-flow, low-gradient AS	4.22	1.57–11.31	0.004	4.35	1.49–12.71	0.007
LVMi—g/m^2^	1.01	1.00–1.02	0.250			
DAPT	1.03	0.31–3.46	0.959			
Pre-procedural Characteristics						
Balloon reimpaction	1.41	0.48–4.13	0.528			
Significant PVL	1.56	0.58–4.19	0.373			
Post-procedural Characteristics						
Hb—g/dL	0.62	0.43–0.89	0.009	0.68	0.43–1.07	0.093
CT-ADP > 180—s	32.73	11.18–95.83	<0.0001	28.93	9.74–85.95	<0.0001
PRI-VASP	0.99	0.96–1.03	0.720			
Mean aortic gradient—mmHg	1.07	1.00–1.14	0.066			
Significant PVL	1.47	0.61–3.55	0.390			
Major vascular complications	3.46	1.37–8.72	0.009	3.01	1.12–8.10	0.029
Echocardiographic Characteristics 1 mo after TAVR						
Mean aortic gradient—mmHg	1.06	0.99–1.15	0.115			
Significant PVL	0.39	0.15–1.06	0.066			
Antithrombotic Regimen after TAVR						
DAPT	1.64	0.56–4.79	0.369			
DOAC	0.91	0.40–2.08	0.826			
VKA	1.17	0.51–2.66	0.716			

ADP = adenosine diphosphate; AF = atrial fibrillation; AS: aortic stenosis; BMI = Body Mass Index; CKD = chronic kidney disease (creatinine > 150mol/L); COPD = chronic obstructive pulmonary disease; CRP = C-reactive protein; CT = closure time; CT-ADP = closure time with ADP; DAPT = dual-antiplatelet therapy; DOAC = direct oral anticoagulants; EuroSCORE = logistic EuroSCORE predicted risk of mortality at 30 d; Hb = hemoglobin level; HR = hazards ratio; PVL = paravalvular leak; LVMi = left ventricular mass index; MLBCs = major or life-threatening bleeding complications; PRI-VASP = platelet reactivity index VASP; SHR = sub-hazards ratio; STS mortality score = The Society of Thoracic Surgery risk score predicted risk of mortality at 30 d; TAVR = transcatheter aortic valve replacement; WBC = white blood cells; VKA = vitamin K antagonist.

**Table 5 jcm-11-00212-t005:** Predictors of post-procedural CT-ADP > 180 s.

		Univariate Analysis			Multivariate Analysis		
		HR	95% CI	*p*-Value	HR	95% CI	*p*-Value
Baseline characteristics							
Hypertension		2.74	0.95–7.87	0.06			
CKD		1.36	0.71–2.60	0.351			
Dialysis		4.04	1.10–14.79	0.035	6.45	1.05–39.49	0.044
Coronary artery disease		1.93	1.09–3.41	0.025	1.79	0.82–3.93	0.145
Stroke history		1.27	0.62–2.60	0.524			
Peripheral artery disease		0.53	0.26–1.06	0.073			
Bleeding history		1.32	0.58–3.01	0.509			
Paradoxical AS		1.48	0.53–4.10	0.456			
LVMi		1.01	1.00–1.02	0.013	1.01	1.00–1.02	0.033
Hb—g/dL		0.81	0.69–0.95	0.012	0.76	0.62–0.95	0.013
Platelet count—/mm^3^		0.99	0.99–1.00	0.001	0.99	0.98–1.00	0.002
DAPT		2.38	1.14–4.94	0.020	1.75	0.60–5.07	0.305
Per procedural characteristics							
Type of valve	SAPIEN	1.04	0.58–1.85	0.903			
	Core Valve	1.05	0.59–1.87	0.863			
Post-procedural characteristics before discharge							
PRI-VASP		0.99	0.97–1.01	0.189			
Mean aortic gradient		1.06	1.00–1.13	0.045	1.04	0.97–1.12	0.290
Significant PVL		2.94	1.61–5.37	<0.0001	2.74	1.26–5.94	0.011

Abbreviations as in Table 4.

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
