# Peer review of "Impact of Primary Hemostasis Disorders on Late Major Bleeding Events among Anticoagulated Atrial Fibrillation Patients Treated by TAVR"

_jcm, 2021, doi:10.3390/jcm11010212_

Round 1

Reviewer 1 Report

The authors of the manuscript focused on the relationship between primary hemostasis disorder and severe bleeding in patients on anticoagulant therapy for atrial fibrilation, treated by TAVR. Patients with atrial fibrillation  treated with oral anticoagulants may be exposed to an increased risk of bleeding events. Anticoagulant therapy is often refrained from out of fear of hemorrhagic complications. The most frequent type of major bleeding is gastrointestinal, but intracranial hemorrhage has the worst prognosis.

This is a very interesting manuscript with a high impact for research in this area. Some parts of the manuscript need to be corrected and supplemented in order for this manuscript to be published.

The methodical part is  precisely written.

Page 2, It is very important that the authors describe the function about VWF in more detail. VWF represents a high-molecular-weight adhesive glycoprotein that plays an essential role in the primary hemostasis by promoting platelet adhesion to the subendothelium and platelet plug formation at the sites of vascular injury. At the same time, the authors should cite the manuscript in which it was stated ,, Semin Thromb Hemost. 2017 Sep;43(6):639-641. doi: 10.1055/s-0037-1603362.

Figures and tables in the text are very clearly written.

I have to say that with these 29 references of which 16 references are in the last 5 years.

Reviewer 2 Report

The authors descibe a crucial aspect of postoperative course of TAVR patients. The association between oral anticoagulation is an actual field of interst.

Reviewer 3 Report

Interesting article
- Why is CT-ADP rarely adopted in clinical practice?
- What advantage does PFA-100® offer over thromboelastography? Are there any concordances? How much does the platelet count affect (these are somewhat generic questions, but I would like these aspects to be minimally mentioned in the introduction and discussion to give more clinical value to the work, which is of high quality).
- However high number of patients despite the excluded patients. Why was CT-ADP not routinely measured in all patients?
-. the HR value / Confidence interval / p-value of the renal insufficiency in the multivariate table 4. Something is wrong, I think it is the p-value or the lower value of the confidence interval (negative?).
- Interesting that PVL affects bleeding in some way
- KM in SPSS must be graphically enhanced, it can be improved. Interesting graph.
- Emphasize in the discussion the concept that therapy must be tailored to the patient. CT-ADP can be a way to tailor post-TAVI therapy
- Excellent article, it should only be enhanced a little from a clinical point of view.
- In this regard, I would like to quote this article on a bleeding complication. In this case the use of thromboelastography and the percentage of ADP inhibition were important. If you can mention it generically in the introduction (DOI: 10.14744 / AnatolJCardiol.2019.78546)

Round 2

Reviewer 1 Report

The presented manuscript has been corrected in response to the suggestions. The authors have followed the recommendations of the reviewer. After the revision, the provided data and addition of the results became more clear. I would like to thank the authors for resubmitting the manuscript and explaining the obscure points from the previous version.